# Protocol: Benefits and harms of remdesivir for COVID-19 in adults: A systematic review with meta-analysis

Asger Sand Paludan-Müller [1,2]*, Andreas Lundh [1,2,3], Matthew J. Page[4], Klaus Munkholm[1,2]

**1** Department of Clinical Research, Centre for Evidence-Based Medicine Odense (CEBMO) and Cochrane Denmark, University of Southern Denmark, Odense, Denmark, **2** Open Patient Data Exploratory Network (OPEN), Odense University Hospital, Odense, Denmark, **3** Department of Infectious Diseases, Hvidovre University Hospital, Hvidovre, Denmark, **4** School of Public Health and Preventive Medicine, Monash University, Melbourne, Australia

* apaludanmuller@health.sdu.dk

**Funding:** ASP-M, AL and KM are supported by Cochrane Denmark and the Centre for Evidence-BasedMedicine Odense (CEBMO). MJP is supported by an Australian Research Council

## Abstract

### Background

Effective drug treatments for Covid-19 are needed to decrease morbidity and mortality for the individual and to alleviate pressure on health care systems. Remdesivir showed promising results in early randomised trials but subsequently a large publicly funded trial has shown less favourable results and the evidence is interpreted differently in clinical guidelines. Systematic reviews of remdesivir have been published, but none have systematically searched for unpublished data, including regulatory documents, and assessed the risk of bias due to missing evidence.

### Methods

We will conduct a systematic review of randomised trials comparing remdesivir to placebo or standard of care in any setting. We will include trials regardless of the severity of disease and we will include trials examining remdesivir for indications other than Covid-19 for harms analyses. We will search websites of regulatory agencies, trial registries, bibliographic databases, preprint servers and contact trial sponsors to obtain all available data, including unpublished clinical data, for all eligible trials. Our primary outcomes will be all-cause mortality and serious adverse events. Our secondary outcomes will be length of hospital stay, time to death, severe disease, and adverse events. We will assess the risk of bias using the Cochranes Risk of Bias 2 tool and the risk of bias due to missing evidence (e.g. publication bias, selective reporting bias) using the ROB-ME tool. Where appropriate we will synthesise study results by conducting random-effects meta-analysis. We will present our findings in a Summary of Findings table and rate the certainty of the evidence using the GRADE approach.

### Discussion

By conducting a comprehensive systematic review including unpublished data (where available), we expect to be able to provide valuable information for patients and clinicians about

Discovery Early Career Researcher Award (DE200101618). The authors received no specific funding for this work. The funders had no role in study design, data collection and analysis, decision to publish, or preparation of the manuscript.

**Competing interests:** The authors have declared that no competing interests exist.

the benefits and harms of remdesivir for the treatment of Covid-19. This will help to ensure optimal treatment for individual patients and optimal utilisation of health care resources.

## Systematic review registration

CRD42021255915.

## Introduction

### Rationale

**Description of the condition.** Covid-19, a disease caused by the severe acute respiratory syndrome coronavirus 2 (SARS-CoV-2) [1], was declared a global pandemic by the WHO in March 11[th], 2020 [2].The pathophysiology of Covid-19 is complex, and the majority of infected people are either asymptomatic or have mild influenza-like illness clinically indistinguishable from other upper respiratory tract infections [3]. However, for some patients the course of the disease is much more serious with involvement of the lower respiratory tract and some patients develop acute respiratory distress syndrome (ARDS) [3]. The infection fatality rate (IFR) of Covid-19 has been the subject of debate and seems to be highly dependent on factors such as age, socio-economic status and pre-existing medical conditions [4]. In October 2020, a report from Imperial College London estimated a IFR in a typical high income countries of 1.15% (95% prediction interval range: 0.78–1.79) [5] whereas a review of 61 seroprevalence studies found a median IFR of 0.27% [4]. As of February 2021 the global death toll due to Covid-19 is estimated at 2,430,640 [6].

**Description of the intervention.** Remdesivir is a nucleotide inhibitor developed by Gilead Sciences, developed initially through research programs in hepatitis C and respiratory syncytial virus (RSV) (https://www.gilead.com/-/media/gilead-corporate/files/pdfs/covid-19/gilead_rdv-development-fact-sheet-2020.pdf) and later through collaboration between Gilead Sciences and the US government seeking to identify therapeutic agents for treating RNA-viruses [7]. Remdesivir has been tested as a treatment for Ebola, SARS, and Middle East Respiratory Syndrome (MERS) and when SARS-CoV-2 was discovered the drug was tested for this condition [7]. In 2020, remdesivir, under the tradename Veklury, was granted conditional marketing authorization for the treatment of Covid-19 by, amongst others, the Federal Drug Administration (FDA) in the United States, the European Medicines Agency (EMA) in the European Union, and Health Canada [8–10]. In the United States remdesivir is approved for treatment of suspected or confirmed Covid-19 in patients 12 years of age or older requiring hospitalization [8], while in the European Union and Canada the drug is approved for the treatment of Covid-19 in patients 12 years of age or older with pneumonia requiring supplemental oxygen [9, 10]. Remdesivir is administered intravenously; in the European Union the EMA approved remdesivir to be given as an initial dose of 200mg on the first day followed by five to ten days of 100mg [11].

**How the intervention might work.** Remdesivir is a prodrug that is metabolized into its active derivative, an adenosine nucleoside triphosphate. Its mechanism of action involves interference with the action of viral RNA-dependent RNA polymerase, resulting in a decrease in viral RNA production; remdesivir has been shown to inhibit replication of SARS-CoV-2 in vitro [7]. Remdesivir has also been shown to reduce levels of virus in the lungs and the amount of lung damage in primates inoculated with the Middle East Respiratory Syndrome Coronavirus (MERS-CoV) [12].

**Why is it important to do this review?.**   Covid-19 has a major impact on individuals, healthcare systems and societies worldwide. While most people with Covid-19 recover from the disease without needing hospitalisation, some develop a more serious course of illness, requiring hospital treatment, which may involve critical care. A high number of patients hospitalized with Covid-19 can substantially overwhelm even very efficient health care systems, resulting in increased mortality in populations, both due to Covid-19 directly and because of suboptimal treatment of other illnesses. Thus, it is important to identify treatments that can reduce the mortality and morbidity of Covid-19. Interim results from an early trial of remdesivir showed promising effects on length of hospitalization and a trend towards an effect on mortality [12]; however, subsequent data from the large WHO funded Solidarity trial has called these effects into question [13].

The evidence regarding the benefits and harms of remdesivir is interpreted differently in various clinical guidelines, with e.g. the WHO conditionally recommending against the use of remdesivir in hospitalised patients [14] and the US National Institutes of Health (NIH) recommending its use in hospitalised patients requiring oxygen who are not receiving mechanical ventilation or ECMO [15].

Among published systematic reviews of trials of remdesivir for Covid-19 [16–21] most did not include data from the largest trial the WHO Solidarity trial [13] and are thus not up to date [16, 18, 19, 22]. Furthermore, none of the reviews systematically acquired data from drug regulators or used a comprehensive method for assessing the potential impact of missing evidence (i.e., trials or trial results) on results of syntheses. Multiple network meta-analyses including remdesivir exist but they are limited by insufficient reporting of methods for assessing transitivity and consistency [16, 17, 19].

Previous systematic reviews have primarily relied on published data as an information source. While all trials that we are aware of have been published a in a biomedical journal, studies have shown that journal publications are not always a reliable source of data for systematic reviews, especially for patient-relevant benefit and harm outcomes [23–25]. Therefore, we will rigorously attempt to identify and acquire all available data for the included trials.

Given the limitations in the extant literature, a comprehensive systematic review including all available data that examines the impact of missing data and reporting bias is needed to provide the best available estimates of benefits and harms of the treatment.

## Objective

To assess the benefits and harms of remdesivir for Covid-19 in adults.

## Methods

### Criteria for considering studies for this review

**Types of studies.**   We will only include randomised trials, including trials using an adaptive design. We will only include the results from the first randomized period of cross-over trials, as the illness is not constant over time and carry over effects of remdesivir are likely. We will include cluster-randomized trials if they have been analysed correctly (e.g. the authors did not conduct a unit of analysis error).

**Types of participants.**   We will include trials of adults (18 years or older) of either sex with any type of comorbidity.

For the assessment of benefits, we will include participants with suspected, probable, or confirmed Covid-19, according to WHO case definitions. We will include participants regardless of severity of disease.

For the assessment of harms, we will include trials including participants whether healthy or with any diagnosis, e.g. Ebola and Middle East respiratory syndrome-related coronavirus.

We will include trials regardless of setting, i.e., participants treated in both in- and out-patient settings.

**Types of interventions.** *Experimental intervention*. Remdesivir. We will consider any dose and mode of administration.

*Comparator intervention*. We will include the following comparators:

- Placebo

- Usual care

*Co-interventions*. We will include trials regardless of whether remdesivir was administered as monotherapy or in combination with other pharmaceutical or non-pharmaceutical interventions provided the co-intervention is delivered to both the intervention and the comparator groups.

Types of outcome measures. *Primary outcomes*

- Overall mortality, defined as the mortality rate from all causes of death in the respective groups.

- Serious adverse events, defined as the number of participants with at least one medical event that is life threatening, results in death, disability, or loss of function, causes hospital admission or prolonged hospitalization. (https://www.ich.org/fileadmin/Public_Web_Site/ICH_Products/Guidelines/Efficacy/E6/E6_R1_Guideline.pdf)

Secondary outcomes

- Length of hospital stay, defined as time to discharge, analysed as a time-to-event outcome.

- Time to death, analysed as time-to-event outcome.

- Number of participants requiring one or more hospitalizations.

- Number of participants with a WHO Clinical Progression Score of 7 or above (i.e. mechanical ventilation, with support in the form of vasopressors, dialysis or extracorporeal membrane oxygenation (ECMO), OR death [26])

- Number of adverse events. We will categorize adverse events according to the classification outlined in the Medical Dictionary for Regulatory Activities terminology (https://www.meddra.org/). We will group events at the system organ class and preferred term level. We will, additionally, report the number of events of all individual adverse effects reported.

- Number of participants with at least one adverse event. We will categorize this outcome as described above.

Trials will be included regardless of whether they report on the outcomes described above. Trials that did not measure these outcomes or for which we were not able to obtain the data will be included in the review and we will summarise the trial characteristics but will not be able to include them in any analyses.

*Timing of outcome assessment*. We will analyse outcomes at day 28 and day 60, defined as the time after randomisation. Where outcomes were assessed at other time points, we will select the outcome closest to (prioritising timepoints after) day 28 or 60.

## Search methods for identification of studies

We will identify eligible trials through various sources:

1) Drug regulatory agencies

   a) We will search for available data from the EMA, FDA, Health Canada, and the Therapeutics Goods Administration (TGA) in Australia to identify potentially eligible trials. We will look at public assessment reports (PARs), review documents, and websites releasing clinical data such as the EMA Clinical Data Website (https://clinicaldata.ema.europa.eu/web/cdp/home) and Health Canadas Clinical Information portal (https://clinical-information.canada.ca/search/ci-rc).

   b) For all eligible trials we will seek to obtain all clinical data submitted by Gilead Sciences to regulatory authorities. The EMA and Health Canada are prospectively releasing clinical data using the portals described above. The FDA and TGA are not routinely releasing all clinical data, but we will submit Freedom of Information (FoI) requests for Clinical Study Reports and other data to both agencies.

2) Trial registries

   a) We will search international trial registries via the WHO International Clinical Trials Registry Platform (ICTRP) for reports of the included trials. The ICTRP platform accepts trials from all major trial registries, including clinicaltrials.gov and the EU Clinical Trials Register (EU-CTR).

   b) We will search clinicaltrials.gov to identify trials that have not yet been added to the ICTRP.

   c) We will also download data from the Covid-19 TrialsTracker (https://covid19.trialstracker.net/) developed by The Datalab and Center for Evidence-Based Medicine from University of Oxford. We will then filter the data for all trials testing remdesivir.

   d) We will also search the Covid-NMA initiative (https://covid-nma.com).

3) Pharmaceutical companies

   a) We will contact Gilead Sciences and ask for all available clinical data, protocols, and statistical analysis plans from randomised trials of remdesivir for any indication.

4) Bibliographic and other databases
   We will search the following databases:

   a) PubMed

   b) Embase

   c) Cochrane Covid-19 Study Register (https://covid-19.cochrane.org).

   d) bioRxiv (https://www.biorxiv.org). Free online preprint archive server. Contains a curated database of COVID-19 and SARS-COV-2 preprints from medRxiv and bioRxiv.

   e) OSF preprints (https://osf.io/preprints/). Free online preprint archive server.

   f) WHO database of Covid-19 publications (https://search.bvsalud.org/global-literature-on-novel-coronavirus-2019-ncov/).

   We will apply very broad search criteria to prioritize sensitivity. The search strategy for all databases will use the following terms: Remdesivir OR Veklury OR GS-5734. We will not apply any restrictions on date or language. If the date of our last search for studies is more than 3 months old at the time of manuscript submission, we will rerun all searches to identify and incorporate any new potentially eligible studies.

5) Contact with trial authors.

We will contact all trial authors by e-mail to obtain a copy of the trial protocol, including any amendments and the statistical analysis plan, and any data or information needed to assess eligibility, calculate effect sizes (i.e. complete datasets and IPD), and perform risk of bias assessment. We will send a second request if we have not received a reply after two weeks.

## Data collection and analysis

**Selection of studies.**   One researcher (ASP) will download trial entries from trial registries, from the Covid-19 TrialsTracker and Covid-NMA and contact Gilead Sciences. One researcher (KM) will search publication databases and download results. One researcher (ASP) will obtain data from regulatory authorities. Two researchers (ASP and KM) will independently screen titles and abstracts of identified reports. One researcher (ASP) will then match publications, trial registry entries and unpublished clinical data for individual trials, so we have a complete set of records for each trial. Two researchers (ASP and KM) will independently assess all full text documents acquired for each trial to determine eligibility of the trial. Disagreements will be resolved by discussion. We will record the study selection procedure in a PRISMA flow diagram.

**Data extraction and management.**   All reports pertaining to individual eligible trials will be linked together in a database.

Two researchers will independently extract data from all documents for the included trials using a MS Excel data collection form, which will be piloted on at least one trial included in the review. Disagreements, will be resolved by discussion, or by consultation with a third review author. If data for an outcome is available from multiple sources, we will employ the following hierarchy: clinical study reports, unpublished clinical data obtained from regulators or investigators, results from trial registry entries, and finally journal publications.

We will extract the following trial, intervention, and population characteristics from each trial:

*Trial meta-data.* Start and completion date of trial, year of publication of report, number of study centres and location, sponsor, funders, details on industry involvement, author last names, author conflicts of interest, source of information (publication, manufacturer, registry, unpublished clinical data).

*Trial methodology.* Design, setting, inclusion criteria, exclusion criteria, total duration of the trial, duration of follow-up, frequency threshold for reporting adverse events (e.g., whether only events occurring in more than 5% of participants are reported), and whether adverse events were actively monitored (pre-specified) or spontaneously reported.

*Participants.* N randomised, N lost to follow-up/withdrawn with reasons, N analysed for each outcome, baseline disease severity classified using modified WHO categories as done elsewhere [27], sex, race, mean age, diagnostic criteria used, and proportion of patients with comorbidities.

*Interventions.* Dosage of remdesivir, comparison, whether the drug was the investigational drug or comparator, concomitant pharmaceutical intervention, and concomitant non-pharmaceutical intervention.

*Outcomes.* Outcome measures of interest (see above) for all timepoints reported. For all trials we will try to obtain individual patient data (IPD). If trials report data stratified by disease severity or duration of symptoms at baseline and randomisation was stratified according to these strata, we will extract both these and aggregate data. Where effect estimates (e.g. mean differences, risk ratios) are extracted, rather than summary statistics (e.g. means and standard deviations), we will extract information on the statistical methods used to obtain the effect estimates.

## Assessment of risk of bias in the included studies

We will use Cochrane's Risk of Bias 2 tool as described in the Cochrane Handbook for Systematic Reviews of Interventions [28] to assess all trials regardless of the nature of the reports of the trial (e.g., CSR or publication). Two researchers will independently assess the risk of bias for all included results. We will resolve any disagreement by discussion or by consultation with a third review author. We will assess the risk of bias according to the following domains:

- Bias arising from the randomization process,

- Bias due to deviations from intended interventions,

- Bias due to missing outcome data,

- Bias in measurement of the outcome,

- Bias in selection of the reported result.

We will use the signalling questions in the RoB 2 tool and rate each domain as "high risk of bias", "low risk of bias" or "some concerns" and will provide a supporting quotation from the trial report together with a justification for our judgment in the "Risk of bias" table. We will summarise the "Risk of bias" judgements across different trials for each of the domains listed. We will, additionally, summarise the overall"Risk of bias" within each trial across domains and for each outcome as low, some concerns or high risk of bias following the approach outlined in Table 8.2.b in the Cochrane Handbook for Systematic Reviews of Interventions [28]; the overall risk of bias within the trial will be the least favourable assessment across the domains, however, where a trial is judged to have some concerns for multiple domains, we will consider rating the overall risk of bias as high. Where necessary, we will contact the trial authors for further information. In the cases where information on the risk of bias was obtained from unpublished data or via correspondence with a trial author, we will note this in the risk of bias table. As adaptive designs may introduce bias in several ways [29], we will consider issues relating to the conduct, analysis and reporting of such trials, when assessing their risk of bias. We will present all "Risk of bias" data graphically and in the text.

## Measures of treatment effect

We will use the freely available software R (https://www.r-project.org/) for all analyses.

**Continuous data.**   As we will handle length of hospitalisation as a time to event outcome, we will not include any continuous outcome data in the review.

**Dichotomous data.**   We will analyse dichotomous data by calculating a pooled odds ratio (OR) with 95% CIs.

**Time to event data.**   For trials where we have acquired access to IPD we will recalculate log hazard ratios (HR) and standard errors (SE) using Cox regression. For trials where IPD is not available but where HRs and a measure of variability is reported we will include this.

For trials where neither IPD nor HRs are available, we will use methods to generate the observed minus expected events (O-E) and the variance (V) as suggested by Tierney et al. [30]. We will choose the appropriate methods based on what is reported for each individual trial.

For trials for which we can obtain HRs and measures of uncertainty, we will include these data in an inverse-variance random effects meta-analysis.

**Unit of analysis issues.**   For studies with multiple arms of the same intervention, we will combine the treatment arms if they can be regarded as providing subtypes of the same treatment and their effect can therefore be considered similar (e.g., different doses within the range of approved dosages or different treatment schedules) to create a single pair-wise comparison

as recommended in the Cochrane Handbook for Systematic Reviews of Interventions [31]. When this is not the case, we will treat each intervention arm separately and will divide out the number of events and total number of participants of the comparator arm approximately evenly among intervention groups.

*Dealing with missing data.* We will contact the primary author or the sponsor to request any missing numerical data and to verify key trial characteristics where necessary, e.g., when data were obtained only from a conference abstract or a clinical trial registry record. We will document details regarding all correspondence. If SDs are not available from the authors, we will calculate the SDs from other available data, if these are reported in the trials, using methods outlined in the Cochrane Handbook for Systematic Reviews of Interventions [28]. If this is not possible, we will substitute SDs with those reported in other similar studies included in the review [32], we will explore the influence of imputing SDs in sensitivity analysis.

*Dichotomous data.* Where data is available but was excluded from the analyses for participants because of protocol non-adherence, we will try to obtain the data from the original trial report or by contacting trial authors. Where data can be obtained for non-adherent participants, we will apply an intention-to-treat (ITT) analysis, in which the number of excluded participants is added to the denominator and the number with events is added to the numerator of the arm to which they were randomised [33]. We will consider the exclusion of ineligible participants who are mistakenly randomized to be appropriate only when information about ineligibility was available at randomization and those making the decision regarding exclusion were blind to allocation; otherwise, we will treat the participants similarly to non-adherent participants [33].

For participants with missing dichotomous outcome data we will follow the methods proposed by Higgins et al. [34] to assess the potential impact of missing data on the results. We will, as a reference, conduct an available case analysis (ACA). For our primary analysis, we will conduct an imputed case analysis (ICA), in which we will impute missing data according to the reasons for missingness (ICA-r). We will specify the categorization of reasons based on a pilot assessment of the study methods, in advance of seeing the data. We will take the uncertainty of the imputed data into account when calculating standard errors, so that these are not inappropriately reduced, and weight the studies accordingly using the methods outlined by Higgins et al. [34]. When the primary meta-analysis suggests an important effect, we will conduct several sensitivity analyses to assess the risk of bias associated with missing participant data. First, we will calculate the best-case and worst-case scenarios, to provide the likely most extreme limits of the effect estimates compatible with the data. Next, we will conduct several analyses by selecting informative missing odds ratios for the two groups that cover more realistic situations [34], based on information about the reasons for missing data and other relevant information. Lastly, we will evaluate the effects of missing participants on the weights awarded to the studies using the method described by Gamble and Hollis [35].

**Primary and secondary analyses.** For all outcomes we will perform two analyses. The primary analyses will be conventional meta-analyses including all available data from all participants randomised in individual trials. As a secondary, exploratory, analysis we will divide studies that report data by baseline disease severity (using the WHO's categorisation of disease severity [26]) and symptom duration (categorised as either 10 days or less, or longer than 10 days) strata into individual entries in the meta-analyses (i.e., each trial-stratum is included in the meta-analysis as an individual trial). For these analyses, we will only include trials where the randomisation was stratified for baseline disease severity and symptom duration and will interpret the findings of these analyses conservatively.

*Data synthesis.* If the treatments, participants, and the underlying clinical question are similar enough for a meta-analysis to be considered meaningful we will undertake pair-wise meta-analysis. Since we expect that there will be heterogeneity between the trials, we plan to use the

random-effects model for pooling trials, regardless of the degree of statistical heterogeneity. We will use the Hartung-Knapp-Sidik-Jonkman method for estimating the heterogeneity variance as it likely outperforms other estimators in most scenarios and results in more adequate type I error rates than the often used Dersimonian and Laird approach [36, 37], especially when the number of trials is small, which we expect to be the case.

For dichotomous outcomes we will use the Mantel-Haenszel method. For outcomes with events rates below 1%, the effects are small and the groups are balanced, we will use the Yusuf-Peto method [28] but will use the Mantel-Haenszel odds ratio method without continuity correction otherwise.

We will, additionally, calculate prediction intervals to provide a better appreciation of the uncertainty around the effect estimate [38], which may be particularly relevant when the between-study heterogeneity is high [39]. As prediction intervals are strongly based on the assumption of a normal distribution for the effects across studies, and they can therefore be problematic when the number of studies is small, we will only calculate them provided there are 10 or more trials and no clear funnel plot asymmetry.

If quantitative synthesis is not appropriate, we will summarise the identified studies narratively.

*Comparisons*. We will make the following comparisons:

- Remdesivir versus placebo

- Remdesivir versus usual care

*Subgroup analyses and investigation of heterogeneity*. Given the risk of type I errors due to multiple testing issues, we will interpret findings from subgroup analyses conservatively. We will perform the following subgroup:

1) Baseline severity. We will categorise trials according to their inclusion criteria regarding disease severity.

2) Setting. We will divide the trials into either hospitalized or outpatient setting.

3) Harms assessed in different indications. We will categorise the trials according to the indication the drug was studied for.

4) Duration of remdesivir treatment. We will divide the trials according to duration of treatment into categories of 1–5 days, 6–10 days, or longer.

5) Type of usual care. We will determine what usual care constituted in the included trials and group the trials by different usual care regimens.

Any additional analyses will be reported as post-hoc.
*Sensitivity analyses*. We will conduct the following sensitivity analyses:

1) Imputation of missing data using any method. We will remove trials that provide results only after missing data were imputed.

2) The influence of imputing SDs. We will remove trials for which we have imputed missing SDs based on the SD of similar studies.

3) Risk of Bias. We will conduct two sensitivity analyses, one excluding only trials considered as being at high risk of bias and one excluding trials considered as being at high risk of bias or some concerns.

4) As the clinical diagnosis of Covid-19 is not very accurate [40], we will explore the influence of including trials with non-confirmed Covid-19 cases by excluding trials in this population.

5) We will explore the influence of including trials including healthy participants and participants with various diagnoses other than Covid-19 on harms by excluding trials examining other indications than Covid-19.

**Assessment of heterogeneity.**   We will assess heterogeneity by comparing participants, interventions, and outcomes between the included studies. In the case of considerable methodological and clinical heterogeneity, we will not pool the data but instead describe them separately and report the clinical diversity of the studies.

We will evaluate the statistical heterogeneity using the $Chi^2$ test and the $I^2$ statistic. We will consider a $Chi^2$ test P value < 0.10 as an indication of statistically significant heterogeneity. We will use the $I^2$ statistic to quantify statistical heterogeneity and will interpret an $I^2$ estimate equal to or greater than 50% as indication of substantial heterogeneity, when accompanied by a statistically significant $Chi^2$ test [28].

We will, additionally, assess heterogeneity by visual inspection of the overlap of the CIs of individual studies in forest plots.

**Assessment of reporting bias.**   For trials with available protocols, we will compare outcomes in the protocol with outcomes in trial reports. If we cannot retrieve the protocol, we will compare the outcomes in the methods section of the report and clinical trial registry records with the reported results.

We will assess the risk of bias due to missing evidence using the preliminary Risk of Bias due to Missing Evidence (ROB-ME) tool (https://www.riskofbias.info/welcome/rob-me-tool). Two review authors (ASP and MP) will independently use the tool for the main analysis for all outcomes in the review. Disagreements will be resolved through discussion or, if necessary, by involving a third review author.

We will construct an outcome matrix building on the template provided in the ROB-ME guidance and will then assess the within-study and across-studies non-reporting bias using the ROB-ME signalling questions and algorithm.

We will present the assessment of risk of bias due to missing evidence in a graph or table along with a brief justification for each judgment. We will display studies with missing results in forest plots as proposed in the ROB-ME guidance.

As part of the ROB-ME assessment, if we can pool ten or more trials that are not similar in size, we will, explore potential small-study effects for the primary outcomes by visually inspecting funnel plots and test for funnel plot asymmetry using Egger's test.

**Summary of findings table.**   We will create a "Summary of Findings" table using all outcomes included in the review.

We will use the Grading of Recommendations, Assessment, Development and Evaluation (GRADE) approach [41] to assess the certainty of the evidence for all outcomes. In the assessment we will consider the domains of risk of bias, inconsistency, indirectness, imprecision, and risk of publication bias. We will use the ROB-ME assessment to inform the assessment of publication bias, downgrading one level for all syntheses rated as 'some concerns' or 'high risk of bias'. We will grade the overall certainty of the evidence for each outcome as 'high', 'moderate', 'low' or 'very low'. We will prepare a 'Summary of findings' table for the each of the planned comparisons. Two review authors will independently make judgments about the certainty of the evidence and disagreements will be resolved by discussion, potentially involving a third review author. Judgments and their justifications will be documented in the "Summary of Findings" tables.

## Supporting information

**S1 Checklist. PRISMA-P 2015 checklist.**
(DOCX)

## Author Contributions

**Conceptualization:** Asger Sand Paludan-Müller, Klaus Munkholm.

**Methodology:** Asger Sand Paludan-Müller, Andreas Lundh, Matthew J. Page, Klaus Munkholm.

**Supervision:** Klaus Munkholm.

**Writing – original draft:** Asger Sand Paludan-Müller.

**Writing – review & editing:** Asger Sand Paludan-Müller, Andreas Lundh, Matthew J. Page, Klaus Munkholm.

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
