## [Decision Letter · Decision Letter 0]

1 Sep 2021

PONE-D-21-16799

PROTOCOL: Benefits and harms of remdesivir for COVID-19 in adults: a systematic review with meta-analysis

PLOS ONE

Dear Dr. Paludan-Müller,

Thank you for submitting your manuscript to PLOS ONE. After careful consideration, we feel that it has merit but does not fully meet PLOS ONE’s publication criteria as it currently stands. Therefore, we invite you to submit a revised version of the manuscript that addresses the points raised during the review process.

We look forward to receiving your revised manuscript.

Kind regards,

Spyridon N. Papageorgiou, DDS, Dr Med Dent

Academic Editor

PLOS ONE

Journal Requirements:

2. Thank you for stating the following in the Acknowledgments/ Funding Section of your manuscript: 

This study is funded internally by Cochrane Denmark and the Centre for Evidence-Based Medicine Odense (CEBMO). MJP is supported by an Australian Research Council Discovery Early Career Researcher Award (DE200101618).

N/A

N/A

Reviewers' comments:

Reviewer's Responses to Questions

**Comments to the Author**

1. Does the manuscript provide a valid rationale for the proposed study, with clearly identified and justified research questions?

Reviewer #1: Yes

Reviewer #2: Yes

2. Is the protocol technically sound and planned in a manner that will lead to a meaningful outcome and allow testing the stated hypotheses?

Reviewer #1: Yes

Reviewer #2: Partly

3. Is the methodology feasible and described in sufficient detail to allow the work to be replicable?

Reviewer #1: Yes

Reviewer #2: Yes

4. Have the authors described where all data underlying the findings will be made available when the study is complete?

Reviewer #1: Yes

Reviewer #2: Yes

5. Is the manuscript presented in an intelligible fashion and written in standard English?

Reviewer #1: Yes

Reviewer #2: Yes

6. Review Comments to the Author

You may also provide optional suggestions and comments to authors that they might find helpful in planning their study.

Reviewer #1: I have reviewed this as a clinical pharmacologist and physician. It appears to be very clear, comprehensive, and of value. I still have a few comments and questions.

Line 116. I am not aware of cluster-randomized trials of remdesivir, but if large enough they may be informative—see https://training.cochrane.org/handbook/current/chapter-23]

Line 120: ‘of both sexes’ → ‘of either sex’

Line 121: The pragmatic question is whether remdesivir reduces harm in patients with ‘Covid-19,’ but the scientific question is whether it reduce harm in patients with proven Covid-19. It may be that a subsidiary analysis would be helpful to answer the latter question.

Line 123: Harms may be related to an interaction between the drug and the disease, so it may distort estimates of harm to include trials in diseases other than Covid-19. The most serious harms will cause death (which is included in the measure ‘all-cause mortality’).

Lines 132 and 354: A problem that has bedevilled trials in Covid-19 has been the heterogeneity of ‘usual care,’ which has in some circumstances included azithromycin, or dexamethasone, or hydroxychloroquine, or any combination of these. How will you deal with this?

Line 136: What if an intervention (notably ventilation, dexametasone) is delivered to some but not all patients in both treatment and control groups?

Line 146: Length of hospital stay & time to death do depend on disease severity. Will this be taken into account in your analysis, or in grading the quality of the research that you include?

Line 166: UK Medicines and Healthcare products Regulatory Agency will also have looked at data from Gilead.

Line 201: I did not understand the sentence ‘We will rerun all searches if the initial search date is greater than 3 months prior to manuscript publication.’

Lines 203 and 295: Is it known how likely authors are to respond to such requests? How will you deal with authors who do not respond?

Line 230: There is no mention of the statistical methods used to analyse trial results.

Line 305: An ITT analysis is entirely appropriate for efficacy, but may not be appropriate for adverse events.

Line 325: The stratified secondary analyses will be interesting and could be clinically relevant.

Line 334: This section will require careful review by a statistician familiar with the field. Reference 37 warns that ‘Even with the HKSJ method, extra caution is needed when there are = <5 studies of very unequal sizes,’ which may well be the case with this study.

Line 374: The accuracy of diagnosis will depend on the prevalence of Covid-19. Most of the trials will have been conducted at times of and in areas of high prevalence.

Reviewer #2: The manuscript 'PROTOCOL: Benefits and harms of remdesivir for COVID-19 in adults: a systemic review with meta-analysis' authored by Paludan-Müller et al. provides a protocol for a systemic review and meta-analysis of remdesivir.

The authors ask a relevant question as conflicting evidence and different interpretations have lead to heterogenous national recommendations and published meta-analyses to date do not fully succeeded in implementing raw and unpublished data. The authors undeline their aim to include such data in the analysis. However, there are lots of meta-analyses published on remdesivir asking the same question, all of which encountered the same issues for data syntheses: heterogenous trial designs and definitions, problems to get raw data in order to seperate aggregated data and harmonize outcomes, limited possibilities for subgroup group analyses. Prior meta-analyses tried to include raw data as well but failed to retrieve for example data from the SOLIDARITY trial which dramatically limits possibilities for valid subgroup analyses. However, robust subgroup analyses with focus on hospitalized patients at an early stage of disease would be of main interest and an overall mortality benefit over all patient groups is not expectable based on individual trial results and existing meta-analyses. From a clinical point of view, the relevant question is not if remdesivir provides a mortality benefit over all hospitalized patients as the answer is most probably known already. The more important question is if there are subgroups that do benefit from remdesivir treatment which can be supported by evidence. Both theoretical assumptions on antiviral activity and subgroup analyses of individual trials ACTT-1 and SOLIDARITY suggest potential efficacy in patients with low flow supplemental oxygen. Seperate subgroup analyses suggest a more pronounced effect of remdesivir when used at an early COVID-19 stage (shorter duration after symptom onset). If this systematic analysis wants to provide novel insights, a comprehensive subgroup analysis inlcuding raw data from clinical trials (e.g. in SOLIDARITY low flow and high flow supplemental oxygen groups are aggregated) is warranted. In addition, data synthesis is limited by different definitions of clinical outcomes (e.g. time to clinical recovery, time to hospital discharge). Here, an intelligent and clinically valid methodology is required in order to incorporate available data in a systematic approach. Aggregation of related outcomes / composite endpoints might be one option if raw data cannot be provided.

In summary, I would recommend to include a subgroup analysis in the protocol and to document constructive considerations how to realize a robust subgroup analysis taking the heterogeneity of clinical trial designs and the resulting complexity for data syntheses into account. As COVID-19 is caused by an acute viral respiratory infection - clinical efficacy of antivirals at later stages of the disease is not expectable based on our current pathophysiologic knowledge. The authors might also profit from exchange with other Cochrane centers that might be working on this topic to provide a unique approach.

The protocol could include information on how to deal with different treatment durations (5 and 10 days of remdesivir) in the trials and how to deal with combinational treatments including remdesivir (e.g. ACTT-2 remdesivir with baricitinib).

Finally, the second time point chosen for analyses of defined outcomes (day 60 after randomisation) might be too optimistic as most trials do not provide follow up data for 60 days to my knowledge. Alternative time points might be considered in the protocol if the 60d point is not covered by sufficient data.

7. PLOS authors have the option to publish the peer review history of their article (what does this mean?). If published, this will include your full peer review and any attached files.

Reviewer #1: No

Reviewer #2: **Yes: **Jakob J. Malin

---

## [Author Response · Author response to Decision Letter 0]

8 Nov 2021

Dear editor,

We would like to thank you and reviewers for their comprehensive reviews and useful comments, that we believe have improved our protocol. Below are our replies to the individual comments. All line numbers refer to the clean version of the manuscript.

Sincerely yours,

Asger Sand Paludan-Müller, on behalf of all authors

Editorial comments

- Done

2. Thank you for stating the following in the Acknowledgments/ Funding Section of your manuscript: 

This study is funded internally by Cochrane Denmark and the Centre for Evidence-Based Medicine Odense (CEBMO). MJP is supported by an Australian Research Council Discovery Early Career Researcher Award (DE200101618).

N/A

- We apologize for this error. We would like the Funding Statement to read the following:

“ASP-M, AL and KM are supported by Cochrane Denmark and the Centre for Evidence-Based Medicine Odense (CEBMO). MJP is supported by an Australian Research Council Discovery Early Career Researcher Award (DE200101618). The authors received no specific funding for this work. The funders had no role in study design, data collection and analysis, decision to publish, or preparation of the manuscript.”

The funding information has now been removed from the manuscript.

N/A

- Please see above.

- Done

- We have checked all references and found no mistakes.

Reviewers' comments:

Reviewer #1: I have reviewed this as a clinical pharmacologist and physician. It appears to be very clear, comprehensive, and of value. I still have a few comments and questions.

- We thank the reviewer for this comment.

Line 116. I am not aware of cluster-randomized trials of remdesivir, but if large enough they may be informative—see https://training.cochrane.org/handbook/current/chapter-23]

- We agree that cluster-randomized trials of remdesivir are unlikely to exist but could be of value. We have changed the sentence to the following:

“We will include cluster-randomized trials if they have been analysed correctly (e.g. the authors did not conduct a unit of analysis error).”

Line 120: ‘of both sexes’ → ‘of either sex’

- Thank you, we have corrected the sentence

Line 121: The pragmatic question is whether remdesivir reduces harm in patients with ‘Covid-19,’ but the scientific question is whether it reduce harm in patients with proven Covid-19. It may be that a subsidiary analysis would be helpful to answer the latter question.

We thank the reviewer for this comment and agree that there is an important distinction between the two. We believe our sensitivity analysis excluding trials where Covid-19 cases are not proven (described in line 385), will help determine whether the answers to the two questions outlined above differ.

Line 123: Harms may be related to an interaction between the drug and the disease, so it may distort estimates of harm to include trials in diseases other than Covid-19. The most serious harms will cause death (which is included in the measure ‘all-cause mortality’).

- We agree that including data from trials with remdesivir used for other indications than Covid-19 may potentially distort estimates. This should be weighed against the benefit of additional power obtained by including more trials. We already plan to explore the impact of including trials in other indications than Covid-19 in a sub-group analysis.

Lines 132 and 354: A problem that has bedevilled trials in Covid-19 has been the heterogeneity of ‘usual care,’ which has in some circumstances included azithromycin, or dexamethasone, or hydroxychloroquine, or any combination of these. How will you deal with this?

- We thank the reviewer for this comment. We agree this might be a major cause of clinical heterogeneity and is likely an issue in many reviews including usual care as a comparator. We have, in response to the reviewers’ comment added a subgroup analysis of different usual care regimens (line 374): 

“5) Type of usual care. We will determine what usual care constituted in the included trials and group the trials by different usual care regimens.

We will also make sure to describe differences in usual care in the narrative part of the review and consider this issue in our grading of the certainty of the evidence.

Line 136: What if an intervention (notably ventilation, dexametasone) is delivered to some but not all patients in both treatment and control groups?

We thank thank the reviewer for this comment. We agree this is a potential source of bias, and one we will be attentive of when assessing the risk of bias in the included trials. We will also report such information when describing the characteristics of the included trials.

Line 146: Length of hospital stay & time to death do depend on disease severity. Will this be taken into account in your analysis, or in grading the quality of the research that you include? 

We thank the reviewer for this comment. In our secondary, exploratory analysis (described on line 334-342) we will analyse data according to disease severity and duration of symptoms.

Line 166: UK Medicines and Healthcare products Regulatory Agency will also have looked at data from Gilead.

We thank the reviewer for this comment. We consider it unlikely that the data submitted to the MHRA would differ from that submitted to the EMA.

Line 201: I did not understand the sentence ‘We will rerun all searches if the initial search date is greater than 3 months prior to manuscript publication.’

- We apologise that the sentence was not clear. We have now hanged it to the following: 

” If the date of our last search for studies is more than 3 months old at the time of manuscript submission, we will rerun all searches to identify and incorporate any new potentially eligible studies”

Lines 203 and 295: Is it known how likely authors are to respond to such requests? How will you deal with authors who do not respond?

- We are not aware of data on the likelihood that authors will respond to data requests. If authors do not respond after our second request, we will not take any additional measures to obtain data directly from the authors. 

Line 230: There is no mention of the statistical methods used to analyse trial results.

- We thank the reviewer for this comment. We have added the following (line 248-250):

” Where effect estimates (e.g. mean differences, risk ratios) are extracted, rather than summary statistics (e.g. means and standard deviations), we will extract information on the statistical methods used to obtain the effect estimates.”

Line 305: An ITT analysis is entirely appropriate for efficacy, but may not be appropriate for adverse events.

- We thank the reviewer for this comment. As we are interested in the effect of assignment to intervention, for adverse events (i.e. harms), we will also perform ITT analysis. 

Line 325: The stratified secondary analyses will be interesting and could be clinically relevant.

- We thank the reviewer for this comment. 

Line 334: This section will require careful review by a statistician familiar with the field. Reference 37 warns that ‘Even with the HKSJ method, extra caution is needed when there are = <5 studies of very unequal sizes,’ which may well be the case with this study.

- We agree that there are important limitations to performing random effects meta-analysis when the number of trials is very small, which may be the case for this review. We will be cognisant of this potential limitation when formulating conclusions. 

Line 374: The accuracy of diagnosis will depend on the prevalence of Covid-19. Most of the trials will have been conducted at times of and in areas of high prevalence.

- We thank the reviewer for this comment. The accuracy of clinical, non-laboratory confirmed diagnoses may, as suggested by the reviewer, be higher in areas with high prevalence. 

We will conduct sensitivity analysis excluding trials with non-laboratory confirmed Covid-19 to explore the effect of including such trials in the review. We will also take the time period and area in which the trial was conducted into consideration when grading our certainty in the body of evidence (via the indirectness domain in GRADE).

Reviewer #2: The manuscript 'PROTOCOL: Benefits and harms of remdesivir for COVID-19 in adults: a systemic review with meta-analysis' authored by Paludan-Müller et al. provides a protocol for a systemic review and meta-analysis of remdesivir.

The authors ask a relevant question as conflicting evidence and different interpretations have lead to heterogenous national recommendations and published meta-analyses to date do not fully succeeded in implementing raw and unpublished data. The authors undeline their aim to include such data in the analysis. However, there are lots of meta-analyses published on remdesivir asking the same question, all of which encountered the same issues for data syntheses: heterogenous trial designs and definitions, problems to get raw data in order to seperate aggregated data and harmonize outcomes, limited possibilities for subgroup group analyses. Prior meta-analyses tried to include raw data as well but failed to retrieve for example data from the SOLIDARITY trial which dramatically limits possibilities for valid subgroup analyses. However, robust subgroup analyses with focus on hospitalized patients at an early stage of disease would be of main interest and an overall mortality benefit over all patient groups is not expectable based on individual trial results and existing meta-analyses. From a clinical point of view, the relevant question is not if remdesivir provides a mortality benefit over all hospitalized patients as the answer is most probably known already. The more important question is if there are subgroups that do benefit from remdesivir treatment which can be supported by evidence. Both theoretical assumptions on antiviral activity and subgroup analyses of individual trials ACTT-1 and SOLIDARITY suggest potential efficacy in patients with low flow supplemental oxygen. Seperate subgroup analyses suggest a more pronounced effect of remdesivir when used at an early COVID-19 stage (shorter duration after symptom onset). If this systematic analysis wants to provide novel insights, a comprehensive subgroup analysis inlcuding raw data from clinical trials (e.g. in SOLIDARITY low flow and high flow supplemental oxygen groups are aggregated) is warranted. In addition, data synthesis is limited by different definitions of clinical outcomes (e.g. time to clinical recovery, time to hospital discharge). Here, an intelligent and clinically valid methodology is required in order to incorporate available data in a systematic approach. Aggregation of related outcomes / composite endpoints might be one option if raw data cannot be provided.

In summary, I would recommend to include a subgroup analysis in the protocol and to document constructive considerations how to realize a robust subgroup analysis taking the heterogeneity of clinical trial designs and the resulting complexity for data syntheses into account. As COVID-19 is caused by an acute viral respiratory infection - clinical efficacy of antivirals at later stages of the disease is not expectable based on our current pathophysiologic knowledge. The authors might also profit from exchange with other Cochrane centers that might be working on this topic to provide a unique approach.

- We thank the reviewer for these very valuable and insightful comments. We agree that other reviews have tried to include unpublished data, however to our knowledge no previous reviews have used as comprehensive a method for identifying and retrieving unpublished data as we propose. Additionally, our author team has substantial experience in getting access to- and working with unpublished data including Clinical Study Reports.[1–5] We agree that it is of major importance to obtain raw data from the SOLIDARITY trial and will make any effort to achieve this, although we are not planning individual patient data (IPD) meta-analyses. 

We also agree that it is of clinical relevance to identify any potential subgroups benefiting from remdesivir treatment. We have taken steps to investigate one important potential moderator of effect, in the form of our secondary analysis analysing participants by their baseline disease severity and duration of symptoms. Of course, this analysis is dependent on obtaining the necessary data, which we believe is feasible. However, for many other potential subgroup analyses that are based on individual participant characteristics, IPD analyses would be required, which we are not planning.

The protocol could include information on how to deal with different treatment durations (5 and 10 days of remdesivir) in the trials and how to deal with combinational treatments including remdesivir (e.g. ACTT-2 remdesivir with baricitinib).

- We thank the reviewer for highlighting this. We apologise it was not clear from the protocol and have now amended the relevant section, which now reads:

“For trials with multiple arms of the same intervention, we will combine the treatment arms if they can be regarded as providing subtypes of the same treatment and their effect can therefore be considered similar (e.g., different doses within the range of approved dosages or different treatment schedules) to create a single pair-wise comparison as recommended in the Cochrane Handbook for Systematic Reviews of Interventions.”

A trial of e.g., remdesivir + baricitinib will not be included in the review, as specified in line 137-139.

Finally, the second time point chosen for analyses of defined outcomes (day 60 after randomisation) might be too optimistic as most trials do not provide follow up data for 60 days to my knowledge. Alternative time points might be considered in the protocol if the 60d point is not covered by sufficient data.

We agree it may be unlikely that trials measure outcomes 60 days after randomization. We will therefore assess the outcome at the time point closest to day 60, so we have amended the sentence to read: 

“Where outcomes were assessed at other time points, we will select the outcome closest to (prioritising timepoints after) day 28 or 60.” (line 166-168)

---

## [Editor Report · Decision Letter 1]

12 Nov 2021

PROTOCOL: Benefits and harms of remdesivir for COVID-19 in adults: a systematic review with meta-analysis

PONE-D-21-16799R1

Dear Dr. Paludan-Müller,

We’re pleased to inform you that your manuscript has been judged scientifically suitable for publication and will be formally accepted for publication once it meets all outstanding technical requirements.

Kind regards,

Spyridon N. Papageorgiou, DDS, Dr Med Dent

Academic Editor

PLOS ONE
---

## [Editor Report · Acceptance letter]

16 Nov 2021

PONE-D-21-16799R1 

PROTOCOL: Benefits and harms of remdesivir for COVID-19 in adults: a systematic review with meta-analysis 

Dear Dr. Paludan-Müller:

I'm pleased to inform you that your manuscript has been deemed suitable for publication in PLOS ONE. Congratulations! Your manuscript is now with our production department. 

Kind regards, 

on behalf of

Dr. Spyridon N. Papageorgiou 

Academic Editor

PLOS ONE